# Extraction of the wake induction and angle of attack on rotating wind turbine blades from PIV and CFD results

Iván Herráez[1], Elia Daniele[2], and J. Gerard Schepers[3]

[1]Faculty of Technology, University of Applied Sciences Emden/Leer, Constantiaplatz 4, D-26723, Emden, Germany
[2]Fraunhofer Institute for Wind Energy and Energy System Technology (IWES), Küpkersweg 70, D-26129, Oldenburg, Germany
[3]Energy Research Center of the Netherlands (ECN), Wind Energy Technology, Westerduinweg 3, 1755 LE Petten, The Netherlands

*Correspondence to:* Iván Herráez (ivan.herraez@hs-emden-leer.de)

**Abstract.** The analysis of wind turbine aerodynamics requires accurate information about the axial and tangential wake induction as well as the local angle of attack along the blades. In this work we present a new method for obtaining them conveniently from the velocity field. We apply the method to the New Mexico PIV-dataset and to CFD simulations of the same turbine. This allows to compare experimental and numerical results of the mentioned quantities on a rotating wind turbine. The presented results open up new possibilities for the validation of numerical rotor models.

## 1 Introduction

The wake induction and the angle of attack (AoA) are crucial quantities for the design and analysis of wind turbines. Several methods have been proposed for their estimation (Hansen et al., 1997; Shen et al., 2009), although their application has been restricted to numerical results (Johansen and Sørensen, 2004; Sørensen et al., 2002). The only exception we are aware of is the work of Yang et al. (2011), who applied the method proposed by Shen et al. (2006) to the experimental data set of the MEXICO turbine. The reason for applying those methods almost exclusively to numerical results is that they usually require more input data than typically available from experiments. Hence, the computation of lift and drag blade characteristics from pressure sensors requires the angle of attack to be computed beforehand from simulations (Bechmann et al., 2011; Herráez et al., 2014). The experimental blade characteristics are therefore paradoxically influenced by the uncertainties of the simulations. Furthermore, pressure measurements are usually only available for a few radial positions, so that the obtainable spanwise resolution is very coarse (Herráez et al., 2014; Sørensen et al., 2002). In addition, according to our experience many of the available methods are often rather complicated to employ and automatize. Last but not least, existing methods usually require user-defined input parameters such as control points or correction models for the root and tip (Rahimi et al.). This increases the uncertainty of the calculations.

### Scope and outline

In order to address the mentioned issues, this work aims at providing with a simple method for obtaining the wake induction and the AoA from the velocity field in the rotor plane. The required flow data can be captured experimentally e.g. by means of Particle Image Velocimetry (PIV) or numerically with Computational Fluid Dynamics (CFD) models. The new method is not only straightforward in its application, but also independent from user-defined input parameters. Furthermore, it takes advantage of the high resolution of PIV and CFD results for obtaining detailed spanwise distributions of the searched quantities.

A brief description of the most common methods for the determination of the wake induction and the AoA is presented in Sec. 2.1. The new method is then introduced in Sec. 2.2. Section 3 describes the experimental and numerical data sets used for our calculations. In Sec. 4 the results of applying the proposed method to the mentioned data sets are presented. The conclusions of this work are then summarized in Sec. 5.

## 2   Determination of the wake induction and the angle of attack

The main challenge for determining the local wake induction and the angle of attack is to obtain the local velocity in the rotor plane without blade bound circulation influences. Once the local undisturbed velocity components are known, the axial and tangential induction factors ($a$ and $a'$, respectively) as well as the AoA are computed as:

$$a = 1 - \frac{u_{ax}}{u_\infty} \tag{1}$$

$$a' = -\frac{u_{tan}}{\omega \cdot r} \tag{2}$$

$$AoA = \arctan\left(\frac{u_{ax}}{u_{tan}}\right) - \theta \tag{3}$$

where $u_{ax}$ and $u_{tan}$ are respectively the rotor plane axial and tangential velocity components free of blade induction influences, $u_\infty$ is the freestream velocity and $\theta$ is the sum of the pitch and local twist angles.

### 2.1   Available methods

As seen above, the determination of the wake induction and the angle of attack requires detailed information about the rotor flow without blade induction. A comparison of several methods for obtaining them concluded that three out of four methods were reasonably consistent and reliable (Guntur et al., 2011). In a very recent benchmark study of multiple methods for computing the AoA (including the one presented here), similar results were obtained from all methods except at the tip and root of the blade, where substantial disagreement between some methods was found (Rahimi et al.). A detailed description of all the existing methods is out of the scope of this article, although the reader is referred to Guntur et al. (2011) and Rahimi et al. for a good overview of the different methods. However, the main ideas behind the most common methods are described in the following:

1. **Inverse Blade Element Momentum (BEM) method:** Given the loads along a blade (which can be obtained from experiments or CFD simulations), the BEM theory can be applied as a reverse engineering model for obtaining the wake induction and AoA. A major drawback of this method is that it strongly relies on the basic assumptions of BEM and the corresponding correction models (e.g. for rotational augmentation and tip effects).

2. **Azimuthal Averaging Technique (AAT) by Hansen et al. (1997):** Several equidistant rings upstream and downstream of the rotor plane are used for probing the velocity field at the radial positions of interest. The average velocity of each ring is then calculated. Finally, the velocity in the rotor plane is computed by interpolating the average velocities of all the rings upstream and downstream of the rotor.

3. **Methods proposed by Shen et al. (2006) and Shen et al. (2009):** For each spanwise position of interest, the velocity field is probed at just one control point of the rotor plane. The blade induced velocity at that point is computed using the law of Biot-Savart. For this, the local bound circulation of all blades must be computed beforehand. In the first version of the method, the bound circulation is assumed to be a point vortex and only the velocity and surface pressure fields are required for its calculation (Shen et al., 2006). In the second version of the method, the bound circulation is distributed along the airfoil, what makes it more realistic (Shen et al., 2009). In that case, the wall shear stress is also required for computing accurately the distributed bound circulation in regions with flow separation. Once the blade induced velocity is known, it must be subtracted from the velocity that was previously probed from the control point. As a result, the local velocity free of blade induction influences is obtained.

## 2.2   New method

In our method, the velocities are directly probed at a location of the rotor plane where the induction of the blades is counterbalanced and cancelled out. With axial, uniform inflow, this condition is fulfilled at the bisectrix between two consecutive blades (this applies for both 2-bladed and 3-bladed rotors). This probing location is represented in Fig. 1. Along that line, the flow symmetry causes the downwash of the blade ahead of the bisectrix to counterbalance the upwash of the blade behind it. Hence, the velocities extracted from the rotor plane are just influenced by the wake induction and do not need to be corrected for blade induction influences. The validity of the method is demonstrated analytically in Appendix A.

The difference between the freestream velocity and the probed axial velocity corresponds to the axial wake induction. The probed tangential velocity corresponds to the tangential wake induction with opposite sign. Equations 1 and 2 can therefore be directly applied for computing the induction factors. However, it is worth to keep in mind that the non-uniformity of the rotor flow is not only caused by the bound circulation, but also by the trailed and shed vorticity. For the cases considered in this work, no shed vorticity needs to be taken into account, since the inflow is axial and uniform and the turbine operation is stationary. The trailed vorticity, though, might play a non-negligible role, especially in the tip region. This is the main limitation of the current method, since the probing position of the velocity field is located far apart from the blade and consequently the trailed vorticity effects can not be well captured. The same issue also applies to other well known and commonly accepted methods, like the AAT, where the computed velocity values are indeed azimuthal averages. Therefore, we do not expect our method to

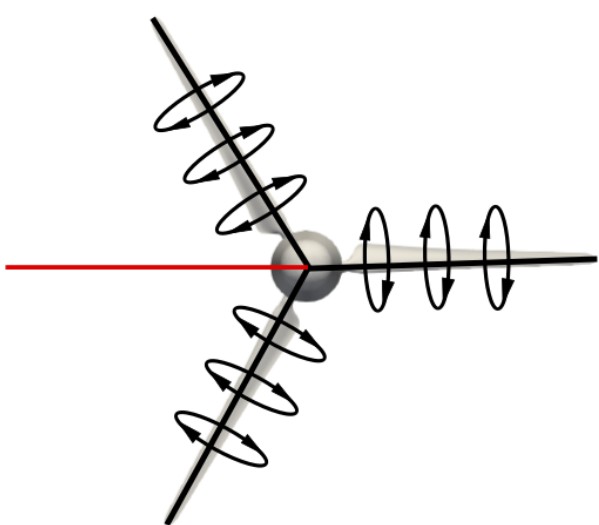

**Figure 1.** Schematic representation of the probing location (red line) of the new method for computing the wake induction and the AoA on a rotor operating under axial, uniform inflow. The bound circulation is represented around the blades. Along the red line, the downwash of the blade located at the 11 o'clock position is cancelled out by the upwash of the blade located at the 6 o'clock position. The blade located at the 3 o'clock position does not induce any velocity along the red line. As a result, the velocities obtained along the red line are free of blade induction influences.

be less accurate than the AAT, although the mentioned limitation should certainly be kept in mind when extracting the AoA from spanwise positions close to the tip. In order to analyse the severity of this limitation, in Sect. 4 our results are not only compared to the AAT, but also to the method proposed by Shen et al. (2006), which presents the advantage that it can take better into account the local influence of the trailing vorticity.

5    In the case of yawed or non-uniform inflow conditions, the new method can still be applied, but it becomes considerably more complex. In that case, the location where the blade induced velocities are counterbalanced must be computed by equalising the induction of all the blades, which must be computed with the law of Biot-Savart. This requires to calculate beforehand the local bound-circulation along each blade. The simplest way to do this is computing the line integral of the velocity field along a closed contour located around the blade section and outside the boundary layer.

## 3    Experimental and numerical data sets

In the following section, the wake induction and the AoA is computed from experimental (PIV) and numerical results of the MEXICO turbine.

PIV windows located just upstream and downstream of the rotor plane are available for the whole blade span (Boorsma and Schepers, 2015). However, a gap of approx. 30 mm exists between both sets of PIV windows, as shown schematically in Fig. 2. Therefore, the data of both measurement sets had to be interpolated in order to obtain the velocities in the rotor plane.

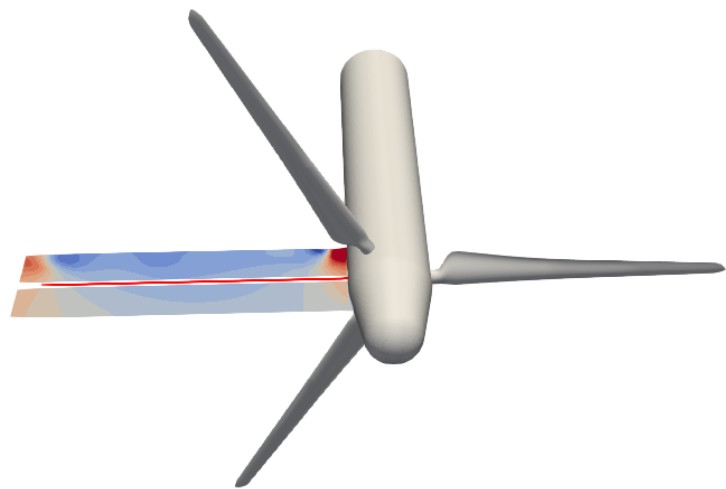

**Figure 2.** Schematic representation of the set of PIV windows located at the 3 o'clock position upstream and downstream of the rotor plane. Between both sets of PIV windows there is a gap of approximately 30 mm, in which the bisectrix between the blades (marked in red) is located. The measurements are phase-locked at $60°$ after blade passage, i.e. when one blade is at the 11 o'clock position (as represented).

The PIV windows were located in a fixed horizontal plane. In this work, we use phase-locked measurements at the azimuth angle $60°$ after blade passage, corresponding to the bisectrix between two blades, as shown in Fig. 1. The available experimental results for each wind speed are in fact the average of 31 image pairs. More information about the experimental set-up can be found in Boorsma and Schepers (2015).

Axial, uniform inflow conditions with three different wind speeds are considered in this work: 10 m/s, 15 m/s and 24 m/s. At 10 m/s the turbine operates in the turbulent wake state, at 15 m/s it is at rated conditions and at 24 m/s it is stalled. The rotational speed of the rotor is kept constant at 424 rpm, what results in a tip speed ratio of $\lambda =10$, 6.67 and 4.7, respectively. The pitch angle is $-2.3°$.

The numerical results used in this work have been extracted from Reynolds Averaged Navier-Stokes simulations of the MEXICO turbine, which were validated against experimental results in Herráez et al. (2014). The reader is referred to that article for a detailed description of the numerical model.

# 4  Results

The results presented in this section have been computed applying the Azimuthal Averaging Technique (AAT) (see Sec. 2.1) to the CFD simulations of the MEXICO turbine as well as applying the new method introduced in Sec. 2.2 to both the CFD and PIV data of the same turbine. Furthermore, experimental AoA results from the method proposed by Shen et al. (2006) have been extracted from Yang et al. (2011) and are also compared here. The application of the AAT to the experimental results was not possible because it requires more data than available from the experiments.

Figure 3 displays the computed axial induction. When the AAT and the new method are applied to the CFD results, convergent results are obtained in the central region of the blade. However, substantial deviations exist in regions with highly three-dimensional flow, i.e. at the root and the tip. At 10 m/s the deviations are stronger at the tip, where $a \geq 0.5$. This large induction factor implies that at least the outer part of the rotor operates under the influence of the turbulent wake state.

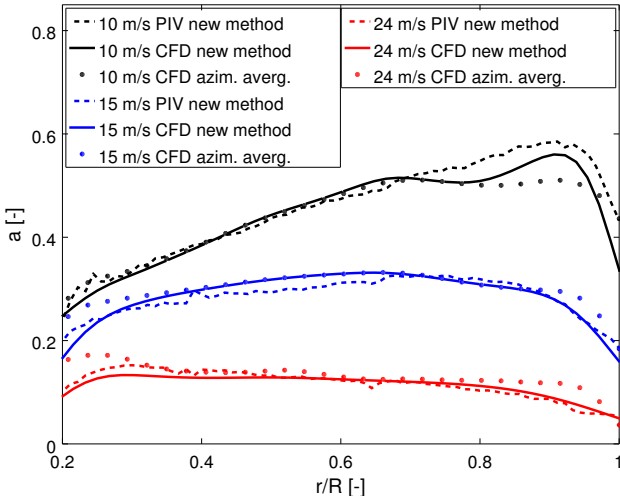

**Figure 3.** Axial induction factor along the rotor radius.

At 15 m/s the disagreement is approximately balanced at the tip and the root. At 24 m/s the deviations are stronger at the root, because of the presence of strong rotational effects (Herráez et al., 2014). It is worth to recall that the AAT method just provides an averaged rotor induction. At the root and the tip, the local induction deviates substantially from the averaged induction as a consequence of the rotor flow non-uniformity, which is attributed to the blade induction and the trailing vorticity influences. The trailing vorticity is not well captured by any of both methods. However, as shown in Sect. 2.2 and demonstrated analytically in Appendix A, the new method accounts correctly for the blade induction influences. This is not the case for AAT, so we consider the results obtained with the new method to be more realistic. This interpretation is supported by the results of two different turbines presented in Rahimi et al., where it is shown that the method proposed by Shen et al. (2006), which

presents the advantage of computing the local blade velocities more accurately, is in excellent agreement with our method in the root and tip region, and it also deviates considerably from the AAT.

The results obtained from the CFD and PIV results using the new method compare reasonably well, although clear discrepancies appear in the tip region at 10 m/s, in the center of the blade at 15 m/s and (to a lower extent) in the root region at 24 m/s. These discrepancies were already expected because of the complex 3-dimensional flows taking place in those regions, as above described. It can be therefore concluded that extracting the wake induction from PIV results opens up new possibilities for the validation of numerical models, since this was until now not possible.

The computed tangential induction factor is displayed in Fig. 4. The results obtained applying the AAT and the new method to the CFD results are in reasonable agreement for outboard positions, where the tangential induction is very low. In the root region, however, substantial discrepancies between both methods appear at 15 and 24 m/s. The same trend is found when the new method is applied to PIV results. As explained above in relation to the axial induction, this behaviour is attributed to the existence of rotational effects in the root region at stall conditions.

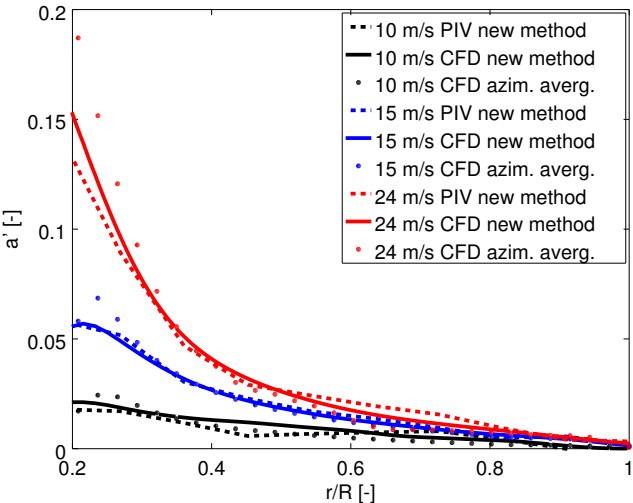

**Figure 4.** Tangential induction factor along the rotor radius.

As shown in Fig. 5, the application of the AAT and the new method to the CFD data provides very similar results for most of the blade span in terms of the AoA distribution, with the exception of the root region at 15 m/s and (more pronouncedly) at 24 m/s. The tangential velocity caused by the blade rotation is directly proportional to the radial position and therefore, in the root region it is comparatively low. This implies that in that region deviations in the prediction of the wake induction have a larger impact on the AoA than at outer radial positions. This is the reason why a very good AoA agreement at outboard positions is obtained in spite of the fact that, as shown in Fig. 3 and 4, the wake induction estimations of both methods are not completely

consistent. The same observation holds true when comparing the AoA obtained from the application of the new method to the PIV and CFD results. The fact that differences in induction are lagely "hidden" in the angle of attack implies that extracting the AoA from CFD simulations and using it for the calculation of experimental 3D polars from pressure sensors is a fairly reliable approach when the AoA can not be directly obtained from experiments.

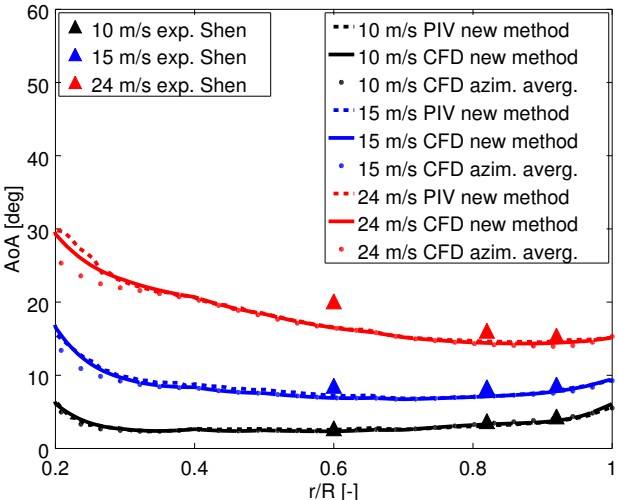

**Figure 5.** Angle of attack distribution along the rotor radius comparing three different methods (new method, azimuthal averaging technique and Shen). The results from Shen have been extracted from Yang et al. (2011).

5     The comparison between the method proposed by Shen et al. (2006) (results obtained from Yang et al., 2011) and the other methods shows a very good agreement along the whole span for operating conditions involving fully attached flow (i.e. $u_\infty = 10\ m/s$). The same happens at the tip ($r/R = 0.92$) for all wind speeds. It is worth to remark that in Yang et al. (2011) several distances between the monitor points and the blade were analysed. The results displayed in Fig. 5 correspond to the smallest distance in order to capture as accurately as possible the trailing vorticity effects (in all cases the distance is at least 2

10  chords, as recommended in Shen et al., 2009). The good agreement between both methods suggests that the tip vortex induction begins to dominate the flow at larger spanwise positions. This is in agreement with Fig. 3.28 of Burton et al. (2011), where it is shown that the induction at the bisectrix between the blades ($\psi = 60°$) is representative for the induction at the blade at least up to $r/R = 0.90$ (at $r/R = 0.97$ this does not hold true any more as a consequence of the tip vortex influences).

    In Fig. 5, the Shen et al. (2006) method presents a slight deviation in the mid-span region with respect to the other methods

15  for $u_\infty = 15\ m/s$ and a substantial deviation for $u_\infty = 24\ m/s$. In this context, it is worth to recall that for $u_\infty = 15\ m/s$, radial flows have been detected in the MEXICO turbine in the mid-span region (see Herráez et al., 2014, Fig. 8b), implying flow separation. For $u_\infty = 24\ m/s$, the radial flows and the flow separation are much more pronounced (see Herráez et al.,

2014, Fig. 9b). Hence, the use of the Kutta-Joukowski theorem for computing the bound circulation, as it is required in Shen et al. (2006), is not expected to be reliable in that case. The reason for this is that the theorem of Kutta-Joukowski only applies to attached flows. This explains the deviations of the Shen et al. (2006) method from the other results in the mid-span region at separated flow conditions. The same applies to a smaller extent at $r/R = 0.82$, where the flow is less separated.

## 5    Conclusions

The method proposed in this work for computing the wake induction and the angle of attack allows to extract accurately those quantities from numerical as well as experimental results of the velocity field in the rotor plane. The method is easy to use and automatize for uniform axial inflow. Furthermore, it does not require any user-defined input parameters. The main limitation of the current method, which also applies to other methods, is that it can not capture the influence of the local trailed and shed vorticities accurately. In principle, this could be a issue at the tip, where the influence of the tip vortex should not be disregarded. However, it has been shown that the method proposed by Shen et al. (2006) (which presents the advantage of capturing the local influences of the trailed and shed vorticities more accurately) provides consistent results with ours up to $r/R = 0.92$, what suggests that the tip vortex induction begins to dominate the flow at larger spanwise positions. For stall conditions, our method has the advantage over Shen's method that it does not rely on the Kutta-Joukowski theorem, which should only be applied to attached flows.

In case of yawed or non-uniform inflow, our method can still be used, but it becomes considerably more complicated because of the dependence of the blade bound circulation on the azimuthal blade position.

CFD simulations have been shown to be capable to predict quite accurately the AoA distribution along the blade span. Therefore, combining the AoA extracted from CFD results with experimental blade pressure measurements can be considered as a reliable approach for obtaining realistic lift and drag blade characteristics. Nevertheless, slight deviations with respect to the purely experimental results should be expected in the root region at stall conditions.

The prediction of the wake induction with CFD is much more challenging than the prediction of the AoA, especially in the tip and root regions. This is especially noticeable at operating conditions involving complex, 3-dimensional flows.

The direct extraction of the wake induction and the AoA from PIV measurements provides valuable information for the validation of numerical models. Another possible application of the new method is the extraction of 100 % experimental blade section characteristics (as long as pressure distributions are also available). This, in turn can, be contribute to the development or enhancement of engineering correction models for different aerodynamic effects.

## Author contribution

I. Herráez proposed the new method for the estimation of the angle of attack and the wake induction, performed the calculations, analysed the results and wrote the manuscript, except the appendix, which was written by E. Daniele. J.G. Schepers and E. Daniele contributed with detailed discussions about the method and the results.

# Appendix A: Counterbalance of the blade induction at the bisectrix between consecutive blades

The new method proposed in this work for computing the wake induction and the angle of attack on the rotor blades of wind turbines is based on the assumption that for axi-symmetric, homogeneous inflow conditions, the induction from the blades bound circulation is counterbalanced at the bisectrix between two blades. In the following, this will be demonstrated for a 3-bladed rotor.

At a given point, the velocity induced by the bound circulation of each blade can be computed using the theorem of Biot-Savart:

$$V(x,y,z) = \frac{1}{4\pi} \int \Gamma(l) \frac{d\boldsymbol{l} \times \boldsymbol{r}}{|\boldsymbol{r}|^3}, \tag{A1}$$

where:

- $V$ is the velocity induced in the point $P = (x, y, z)$.

- $\Gamma$ vorticity distribution function of the position ($Q$) along the blade span-wise direction because of the bound vorticity of a single blade.

- $l$ the coordinate along the blade span-wise direction.

- $r$ the distance between the probe point and the blade element considered.

We introduce two angles:

- $\beta$ is the azimuth angle of the sampling point. It is positive if counter-clock-wise.

- $\psi$ is the azimuth angle of the blade segment. It is positive if counter-clock-wise.

Expressing the quantities in the Cartesian reference frame we have:

$$d\boldsymbol{l} \qquad = (0,\ dl\sin\psi,\ dl\cos\psi) \tag{A2}$$

$$P \qquad = (0,\ r\sin\beta,\ r\cos\beta) \tag{A3}$$

$$Q \qquad = (0,\ l\sin\psi,\ l\cos\psi) \tag{A4}$$

$$\boldsymbol{r} \quad = (0,\ r\sin\beta - dl\sin\psi,\ r\cos\beta - dl\cos\psi) \tag{A5}$$

The cross product in the Cartesian reference frame results in:

$$d\boldsymbol{l} \times \boldsymbol{r} =$$
$$= (dl_y r_z - dl_z r_y)\hat{i}+$$
$$\quad (dl_z r_x - dl_x r_z)\hat{j}+$$
$$\quad (dl_x r_y - dl_y r_x)\hat{k}$$
$$= (dl_y r_z - dl_z r_y)\hat{i}+$$
$$\quad 0\hat{j}+$$
$$\quad 0\hat{k}$$
$$= [dl\sin\psi(r\cos\beta - l\cos\psi) - dl\cos\psi(r\sin\beta - l\sin\psi)]\,\hat{i}+$$
$$\quad 0\hat{j}+$$
$$\quad 0\hat{k} \tag{A6}$$
$$= dl\,[r(\cos\theta\sin\psi - \sin\beta\cos\psi) - l(\cos\psi\sin\psi - \cos\psi\sin\psi)]\,\hat{i}+$$
$$\quad 0\hat{j}+$$
$$\quad 0\hat{k}$$
$$= dl\cdot r(\cos\beta\sin\psi - \sin\beta\cos\psi)\hat{i}+$$
$$\quad 0\hat{j}+$$
$$\quad 0\hat{k}$$
$$= dl\cdot r\sin(\psi - \beta)\hat{i}+$$
$$\quad 0\hat{j}+$$
$$\quad 0\hat{k}.$$

The denominator of the integrand in Equation A1 in the Cartesian reference frame results in:

$$|\boldsymbol{r}|^3 =$$
$$= \left[(r\sin\beta - l\sin\psi)^2 + (r\cos\beta - l\cos\psi)^2\right]^{\frac{3}{2}}$$
$$= \left[r^2\sin^2\beta + l^2\sin^2\psi - 2lr\sin\beta\sin\psi + r^2\cos^2\beta + l^2\cos^2\psi - 2lr\cos\beta\cos\psi\right]^{\frac{3}{2}} \tag{A7}$$
$$= \left[r^2 + l^2 - 2lr\cos(\beta - \psi)\right]^{\frac{3}{2}}.$$

5  Because of the presence of the sole $x$-component of the induced velocity $V$ we write for simplicity the Equation A1 in the Cartesian reference frame as:

$$u(l, r, \beta, \psi) = \frac{1}{4\pi}\int_0^R \Gamma(l)\frac{r\sin(\psi - \beta)}{[r^2 + l^2 - 2lr\cos(\beta - \psi)]^{\frac{3}{2}}}\,dl, \tag{A8}$$

where $R$ is the blade radius. This equation holds for every single blade, $i \in [0, 1, \dots, N-1]$, where $N$ is the number of blades for the considered rotor. The difference in azimuthal angle for each blade is simply given by:

$$\Delta\psi = \frac{2\pi}{N}, \tag{A9}$$

being the azimuthal angle of the $i$-th blade given by:

$$\psi_i = \psi_0 + i\Delta\psi. \tag{A10}$$

Having this, the velocity from the $i$-th blade could be expressed in terms of the 0-th blade azimuthal angle and the relative difference, as follows:

$$u_i = \frac{1}{4\pi} \int_0^R \Gamma_i(l) \frac{r\sin(\psi_0 + i\Delta\psi - \beta)}{[r^2 + l^2 - 2lr\cos(\beta - \psi_0 - i\Delta\psi)]^{\frac{3}{2}}} \, dl, \tag{A11}$$

summed up for all the blades, this provides the velocity induced by the rotor (i.e. all blades together):

$$u_{Rotor} = \sum_{i=0}^{N-1} u_i$$

$$= \sum_{i=0}^{N-1} \frac{1}{4\pi} \int_0^R \Gamma_i(l) \frac{r\sin(\psi_0 + i\Delta\psi - \beta)}{[r^2 + l^2 - 2lr\cos(\beta - \psi_0 - i\Delta\psi)]^{\frac{3}{2}}} \, dl. \tag{A12}$$

At the bisectrix between two consecutive blades, we have:

$$\Delta\psi = \frac{2\pi}{N}, \; \beta = \frac{\pi}{N} = \frac{\Delta\psi}{2}.$$

Equation A12 can therefore be rewritten as:

$$u_{Rotor} = \sum_{i=0}^{N-1} \frac{1}{4\pi} \int_0^R \Gamma_i(l) \frac{\overbrace{r\sin\left(\psi_0 + i\frac{2\pi}{N} - \frac{\pi}{N}\right)}^{A}}{\left[r^2 + l^2 - 2lr\underbrace{\cos\left(\frac{\pi}{N} - \psi_0 - i\frac{2\pi}{N}\right)}_{B}\right]^{\frac{3}{2}}} \, dl. \tag{A13}$$

Considering now a 3-bladed rotor we would have that the numerator $A$ for each single blade becomes:

$$A = \sin\left(\psi_0 + \frac{\pi}{N}(2i - 1)\right),$$
$$A_0 = \sin\left(\psi_0 - \frac{\pi}{N}\right),$$
$$A_1 = \sin\left(\psi_0 + \frac{\pi}{N}\right),$$
$$A_2 = \sin\left(\psi_0 + \frac{3\pi}{N}\right),$$

without loss of generality, we set $\psi_0 = 0$, implying that the first blade is at 12 o'clock position. Hence:

$$A_0 = \sin\left(-\frac{\pi}{3}\right)$$
$$A_1 = \sin\left(+\frac{\pi}{3}\right)$$
$$A_2 = \sin\left(+\frac{3\pi}{3}\right),$$

i.e. $A_0 = -A_1$, $A_2 = 0$. Concerning the term $B$ of Equation A13 a similar procedure leads to:

$$B = \cos\left(-\psi_0 + \frac{\pi}{N}(1 - 2i)\right)$$
$$B_0 = \cos\left(-\psi_0 + \frac{\pi}{N}\right)$$
$$B_1 = \cos\left(-\psi_0 - \frac{\pi}{N}\right)$$
$$B_2 = \cos\left(-\psi_0 - \frac{3\pi}{N}\right),$$

5      and setting $\psi_0 = 0$, we have:

$$B_0 = \cos\left(+\frac{\pi}{3}\right)$$
$$B_1 = \cos\left(-\frac{\pi}{3}\right)$$
$$B_2 = \cos\left(-\frac{3\pi}{3}\right),$$

i.e. $B_0 = B_1$, $B_2 = -1$.

Plotting now the fraction of the integrand of Eq. A13, Fig. A1 is obtained. As it can be seen, for each section along the blade,
10    the velocities must be sample exactly at the bisectrix between two blades (i.e. at $\psi = 60°$, $180°$ or $300°$ ) in order to achieve a
zero blade induction value.

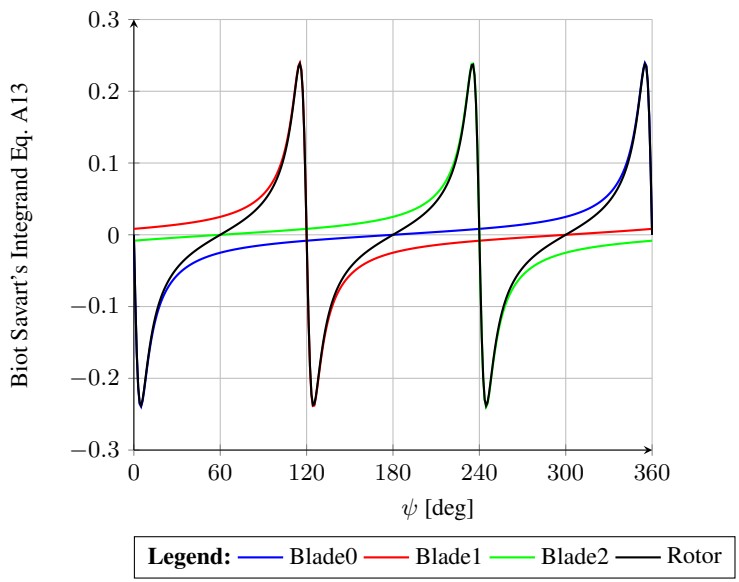

**Figure A1.** Biot-Savart's integrand from Eq. A13 for each blade for a 3-bladed rotor (blades located at $\psi = 0°, 120°$ and $240°$). The sum of all the blade contributions is shown in black. At the bisectrix between two consecutive blades ($\psi = 60°, 180°$ and $300°$), the Biot-Savart's integrand from Eq. A13 is zero. The blade induced velocity at those points is consequently also zero.

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
