# Peer review of "Extraction of the wake induction and angle of attack on rotating wind turbine blades from PIV and CFD results"

_Wind Energy Science, 2017_

## Referee Comment (RC1) · Anonymous Referee #1 · 28 Oct 2017

Review on the manuscript entitled "Brief communication: Extraction of the wake induction and angle of attack on rotating wind turbine blades from PIV and CFD results" by Herraez, Daniele and Schepers

The manuscript presents a method for determining the angle of attack on rotor blades using data from both experimental and CFD computational results. The method uses velocity data probed at the "counterbalanced" locations, i.e. along the "bisectrix" lines between the blades which are located at very long distances (for example at r/R=80% the distance is about 16 chords) from the candidate blade. Since the wake induction has the non-uniformity in the rotor plane due to the wake, the obtained "AOA" cannot

represent the true AOA at which the blade experiences. Thus, this reviewer cannot recommend the manuscript for publication.

Specific comments: 1. Page 1, Line 5: "although their application has been restricted to numerical results". This is not true. From this reviewer's knowledge, Shen et. al., 2009 was used for determining the AOA on rotating blades using the MEXICO measurement data (Wind Energy 2011, Vol. 14(4), pp.539-556). 2. Page 2, Line 14: "u_ax" and "u_tan" are not constant in the rotor plane. "undisturbed" is not appropriate. 3. Line 52: "at just one single arbitrary control point of the rotor plane". This is not true. In the method, one control point was used for each blade cross-section and the selection was not "arbitrary". 4. When PIV data is used, is the velocity data extraction only made when the blades at specific azimuth positions, i.e. when the PIV location is a "bisectrix"? Any averaging was performed?

---

## Referee Comment (RC2) · Anonymous Referee #2 · 28 Oct 2017

Review of paper WES-2017-43

Brief communication: Extraction of wake induction and angle of attack on rotating wind turbine blades from PIV and CFD results.

Authors: I. Herraez et al.

The brief communication presents a new method for determining the blade sectional angle-of-attack distribution from PIV and CFD results. Experimental and computational results are analyzed for the MEXICO rotor (new MEXICO data). Discrepancies between PIV data and CFD results are discussed with emphasis on root and tip effects. While axial-/angular induction factors show reasonable agreement, angle-of-attack distributions agree quite well which is a nice overall result.

General comments:

The paper is written quite well (check for a few typos and replace 'like' by more appropriate choice of word) and is of interest to the wind energy community. The reviewer appreciates the conceptual idea of determining the angle of attack at the bisectrix between blades as it opens an opportunity for a consistent comparison between experiments and computations. The proposed method is easy to implement/apply to any wind turbine rotor. This is the major strength of the manuscript. - However, the reviewer has a few comments of concern, see below:

1. MAJOR CONCERN (has to be addressed with supporting data before manuscript can get published): Is the bisectrix line representative of the angle of attack along the blades ? Both books of Manwell and Burton do have a figure on the azimuthal variation of the induced flow in the rotor plane. Now how can the angle of attack measured at the bisectrix be representative ? This would work if the authors can show that indeed V0(1-a)/[Om*r*(1+a')] is the same along the blades and along the bisectrix lines. Again, the method is interesting but there is no support for its physical correctness. This has to be addressed with some analyses/results before the paper can be accepted! This would benefit the manuscript and make it highly relevant to the wind energy science community.

2. Section 2 would absolutely benefit from a clear sketch illustrating the velocity sampling of past methods and the present method. I think this could be done easily and the manuscript would benefit a lot from a clear illustration.

3. As a follow-up to comment 2, the referee would have to explore the details of Method 3 [Shen 2009] but I don't think the latter is described fully correct (page 3, "arbitrary control point" ?). Again, a clear sketch would probably be of help.

4. As a follow-up to comments 1-3, the new method should be shown to be superior

to older methods. The authors can probably retrieve CFD data to apply to at least a couple of the older methods.

5. The authors state at least four times that the new method "opens up new possibilities". - Which ones ?

---

## Editor Comment (EC1) · J.ÂăN. Sørensen (Editor) · 1 Nov 2017

The reviewer points at two main problems with the paper:

a. It claims originality. However, a similar analysis was carried out using Mexico-rotor data in the paper by Yang et al. 'Extraction of airfoil data using PIV and pressure measurements' from WE 2011. It is required to clearly state the difference between your contribution and the previous paper.

b. The reviewer claims that it is not the AOA that your method computes, as it is based on the velocity at a point far away from the blades. It is therefore needed that you

document that it is indeed the same velocity that you would get at the blade, i.e. the velocity induced by the free vortices corresponding to the one used in BEM analyses.

Besides there is no explanation of how the measurements were collected. Since the blade is rotating, it is unclear how the PIV data are obtained at the bisectrix. Either the PIV-data are taken in a co-rotating system or they are phase-averaged. If the latter is the case, the full time-history should be shown in order to compare the azimuthal dependence with the bisectrix value and the azimuthally averaged values with the full time (azimuth) history.

Please address these problems (plus the remaining ones) properly in your reply.

———————————————————

---

## Editor Comment (EC2) · J.ÂăN. Sørensen (Editor) · 1 Nov 2017

The comments of reviewer 2 are generally in line with those by reviewer 1. But he also states that the issue, if your method really represents the real AOA, needs to be supported by data, before the paper can be accepted for publication. Please address this plus the remaining items pointed out by the reviewer in your reply.
* * *

---

## Author Comment (AC1) · 15 Nov 2017

**Reply to the comments of referee #1 on the manuscript "Brief communication: extraction of the wake induction and angle of attack on rotating wind turbine blades from PIV and CFD results"**

Iván Herráez, Elia Daniele and J. Gerard Schepers

We would like to thank the reviewer for his/her valuable comments and suggestions. We believe that they contributed to improve the quality of the manuscript. In the following, we present our replies.

The new version of the manuscript is appended to this document. Changes to the original manuscript have been highlighted in blue color.

**General comment**

*The method uses velocity data probed at the counterbalanced locations, i.e. along the bisectrix lines between the blades which are located at very long distances (for example at r/R=80% the distance is about 16 chords) from the candidate blade. Since the wake induction has the non-uniformity in the rotor plane due to the wake, the obtained AOA cannot represent the true AOA at which the blade experiences.*

Reply:

As the referee correctly remarks, the non-uniformity of the flow is an important challenge that needs to be considered for obtaining realistic results. The non-uniformity is due to the blade induction, the trailed vorticity and the shed vorticity. The shed vorticity can be disregarded in this case, since we consider only stationary, axi-symmetric, uniform inflow conditions. The blade induction is cancelled out at the bisectrix between the blades, as we demonstrate analytically in the appendix of the new version of the manuscript. The trailed vorticity is admittedly not well captured by the new method and this could be a issue in the tip region, where the non-uniformity is stronger as a consequence of the tip vortex induction. It is worth to remark that this issue also applies to other commonly accepted methods, like the Azimuthal Averaging Technique (AAT), so we do not expect a lower accuracy from our method. In order to investigate the severity of this limitation, we compare now our results to the ones presented in Yang et al. [2011] using Shen et al. [2006]'s method. That method has the advantage that it can compute more accurately the local trailed vorticity induction thanks to the proximity of the monitor point to the blade. The results are presented in Fig. 5 and show a good agreement between both methods in the tip region for all wind speeds. This demonstrates that the blade induction clearly dominates over the trailed vorticity induction at least up to $r/R = 0.92$. This is in agreement with Fig. 3.28 of Burton et al. [2011], where it is shown that the induction at the bisectrix between two blades ($\psi = 60°$) is representative of the local induction at the blade at least up to $r/R = 0.90$. In the mentioned reference it is also shown that for $r/R = 0.97$ this is not true any more, what we attribute to strong tip vortex induction influences.

All in all, we show that our method is reliable at least up to $r/R = 0.92$. For larger radial positions we recommend to use the method proposed by Shen et al. [2006].

Apart from that, our method presents the advantage that it does not rely on the Kutta-Joukowski theorem, what makes it more reliable for stall conditions.

These advantages and drawbacks of the new method with respect to the other methods have now been described in detail in the new version of the manuscript.

We are therefore convinced that the current work can be useful for the scientific community.

**Specific comments**

1. *Page 1, Line 5: "although their application has been restricted to numerical results". This is not true. From this reviewers knowledge, Shen et. al., 2009 was used for determining the AOA on rotating blades using the MEXICO measurement data (Wind Energy 2011, Vol. 14(4), pp.539-556).*
   Reply: **Thanks for this hint, we were not aware of that publication. We mention now that paper in our manuscript and, as discussed above, we have included the AoA results from that paper in Fig. 5.**

2. *Page 2, Line 14: $u_{ax}$ and $u_{tan}$ are not constant in the rotor plane. "undisturbed" is not appropriate.*
   Reply: **"undisturbed" has been changed to "free of blade induction influences".**

3. *Line 52: "at just one single arbitrary control point of the rotor plane". This is not true. In the method, one control point was used for each blade cross-section and the selection was not "arbitrary".*
   Reply: **The sentence was perhaps not well formulated. It stated "For each spanwise position of interest, the velocity field is probed at just one single arbitrary control point of the rotor plane.". We meant precisely that, as the referee says, one control point was used for each blade cross-section. The word "arbitrary" is admittedly not correct, since as stated in Shen et al. [2009], the monitoring point should be located at a certain distance from the blade of about 2 chords (at least). We have now reformulated the sentence for making it more clear.**

4. *When PIV data is used, is the velocity data extraction only made when the blades are at specific azimuth positions, i.e. when the PIV location is a "bisectrix"? Any averaging was performed?*
   Reply: **The velocity data extraction is only performed $60°$ after the blade passage (i.e. at the bisectrix between two consecutive blades). The velocity field measurements are phase-locked and for each PIV window, the available experimental results are averaged over 31 image pairs [Boorsma and Schepers, 2015]. This information has been added to the new version of the manuscript.**

[revised manuscript text omitted]

---

## Author Comment (AC2) · 15 Nov 2017

**Reply to the comments of referee #2 on the manuscript "Brief communication: extraction of the wake induction and angle of attack on rotating wind turbine blades from PIV and CFD results"**

Iván Herráez, Elia Daniele and J. Gerard Schepers

We would like to thank the reviewer for his/her valuable comments and suggestions. We believe that they contributed to improve the quality of the manuscript. In the following, we present our replies.

The new version of the manuscript is appended to this document. Changes to the original manuscript have been highlighted in blue color.

**Comments**

1. *MAJOR CONCERN (has to be addressed with supporting data before manuscript can get published): Is the bisectrix line representative of the angle of attack along the blades? Both books of Manwell and Burton do have a figure on the azimuthal variation of the induced flow in the rotor plane. Now how can the angle of attack measured at the bisectrix be representative ? This would work if the authors can show that indeed $V_0(1-a)/[\omega \cdot r \cdot (1 + a')]$ is the same along the blades and along the bisectrix lines. Again, the method is interesting but there is no support for its physical correctness. This has to be addressed with some analyses/results before the paper can be accepted! This would benefit the manuscript and make it highly relevant to the wind energy science community.*

   **Reply:**

   The azimuthal flow non-uniformity is due to the blade induction, the trailed vorticity and the shed vorticity. The shed vorticity can be disregarded in this case, since we consider only stationary, axi-symmetric, uniform inflow conditions. The blade induction is cancelled out at the bisectrix between the blades, as we demonstrate analytically in the appendix of the new version of the manuscript. The trailed vorticity is admittedly not well captured by the new method and this could be a issue in the tip region, where the non-uniformity is stronger as a consequence of the tip vortex induction. It is worth to remark that this issue also applies to other commonly accepted methods, like the Azimuthal Averaging Technique (AAT), so we do not expect a lower accuracy from our method. In order to investigate the severity of this limitation, we compare now our results to the ones presented in Yang et al. [2011] using Shen et al. [2006]'s method. That method has the advantage that it can compute more accurately the local trailed vorticity induction thanks to the proximity of the monitor point to the blade. The results are presented in Fig. 5 and show a good agreement between both methods in the tip region for all wind speeds. This demonstrates that the blade induction clearly dominates over the trailed vorticity induction at least up to $r/R = 0.92$. This is in agreement with Fig. 3.28 of Burton et al. [2011], where it is shown that the induction at the bisectrix between two blades ($\psi = 60°$) is representative of the local induction at the blade at least up to $r/R = 0.90$. In the mentioned reference it is also shown that for $r/R = 0.97$ this is not true any more, what we attribute to strong tip vortex induction influences.

   All in all, we show that our method is reliable at least up to $r/R = 0.92$. For larger radial positions we recommend to use the method proposed by Shen et al. [2006].

Apart from that, our method presents the advantage that it does not rely on the Kutta-Joukowski theorem, what makes it more reliable for stall conditions.

These advantages and drawbacks of the new method with respect to the other methods have now been described in detail in the new version of the manuscript.

We are therefore convinced that the current work can be useful for the scientific community.

2. *Section 2 would absolutely benefit from a clear sketch illustrating the velocity sampling of past methods and the present method. I think this could be done easily and the manuscript would benefit a lot from a clear illustration.*

   Reply:
   We have included the requested sketch showing the velocity sampling line of new method (Fig. 1) and the location of the corresponding PIV windows (Fig. 2). For the older methods, we refer to Hansen et al. [1997] and Shen et al. [2006].

3. *As a follow-up to comment 2, the referee would have to explore the details of Method 3 [Shen 2009] but I dont think the latter is described fully correct (page 3, "arbitrary control point" ?). Again, a clear sketch would probably be of help.*

   Reply:
   The word "arbitrary" is admittedly not correct. Shen et al. [2009] recommends the point to be located at a distance of 2 chords from the blade. This has been corrected in the new version of the manuscript.

4. *As a follow-up to comments 1-3, the new method should be shown to be superior to older methods. The authors can probably retrieve CFD data to apply to at least a couple of the older methods.*

   Reply:
   Our results are now not only compared to the Azimuthal Averaging Technique [Hansen et al., 1997], but also to the method proposed by Shen et al. [2006] (extracted from Yang et al. [2011]).

   From the presented results, we conclude that the accuracy of the new method is comparable to

   seems to be comparable to previous methods. The new method is superior in terms of simplicity, since it only requires the velocity field and is much easier to implement than the previous methods. Furthermore, it does not require user-defined input-parameters as monitor points or correction models for the tip and root. It also presents the advantage that it does not rely on the Kutta-Joukowski theorem, so in principle it is more reliable for stalled flows.

5. *The authors state at least four times that the new method "opens up new possibilities". - Which ones ?*

   Reply:
   As we state in the introduction, the computation of lift and drag characteristics from pressure sensors requires the angle of attack to be computed beforehand from simulations. This implies that the experimental blade characteristics are influenced by the simulation uncertainties. The new method allows to extract easily the angle of attack from PIV results without using any simulation. This contributes to reduce the uncertainties and the computational costs. Therefore, studying aerodynamic effects involving the comparison between 2D and 3D airfoil characteristics can now be done

100% experimentally and with less uncertainty. Furthermore, up to now numerical simulations could not be validated against experimental results in terms of the wake induction. With the current method this is now also possible. These informations are included in the new version of the manuscript.

[revised manuscript text omitted]

---

## Author Comment (AC3) · 15 Nov 2017

**Reply to the remarks of the editor to the comments of referee #1 on the manuscript "Brief communication: extraction of the wake induction and angle of attack on rotating wind turbine blades from PIV and CFD results"**

Iván Herráez, Elia Daniele and J. Gerard Schepers

**We would like to thank the editor for his complementary remarks to the comments of referee #1. In the following, we present our replies.**

**Comment**

1. *The reviewer points at two main problems with the paper:*

    (a) *It claims originality. However, a similar analysis was carried out using Mexico-rotor data in the paper by Yang et al. "Extraction of airfoil data using PIV and pressure measurements" from WE 2011. It is required to clearly state the difference between your contribution and the previous paper.*

    (b) *The reviewer claims that it is not the AOA that your method computes, as it is based on the velocity at a point far away from the blades. It is therefore needed that you document that it is indeed the same velocity that you would get at the blade, i.e. the velocity induced by the free vortices corresponding to the one used in BEM analyses*

    **Reply:**

    (a) **We were not aware of the paper by Yang et al. [2011]. In the new version of the manuscript we cite it and compare their results to ours. Some important differences between both papers are described in the following:**

    - **In the case of Yang et al. [2011], the calculations of the AoA are based on a method that had been developed 5 years before by Shen et al. [2006]. In our case, we introduce and employ a completely different and new method.**
    - **The method by Shen et al. [2006] requires pressure and PIV measurements. Our method just needs PIV measurements.**
    - **The results from Yang et al. [2011] only show the AoA for 3 radial positions ($r/R = 0.6, 0.82$ and $0.92$). In our case, we present it for the whole blade span from the root until the tip.**
    - **Our paper, in opposition to the one of Yang et al. [2011], not only presents the AoA but also the axial and tangential induction.**
    - **We compare the results from 3 different methods and discuss their advantages and disadvantages, whereas Yang et al. [2011] does not compare the results with other methods.**
    - **We compare the experimental results with CFD results, whereas Yang et al. [2011] use the experimental results for adapting the corresponding 3D airfoil polars and running a BEM model.**

    **We believe that both papers are very useful for the scientific community and complement well each other. The focus of Yang et al. [2011] is the extraction of the airfoil characteristics from experimental PIV and pressure measurements and we rather focus on the presentation and validation of a new method.**

(b) **A detailed discussion of this issue has been included in the new version of the manuscript, including an analytical demonstration in the appendix.**

2. *Besides there is no explanation of how the measurements were collected. Since the blade is rotating, it is unclear how the PIV data are obtained at the bisectrix. Either the PIV-data are taken in a co-rotating system or they are phase-averaged. If the latter is the case, the full time-history should be shown in order to compare the azimuthal dependence with the bisectrix value and the azimuthally averaged values with the full time (azimuth) history*

   **Reply:**

   **As explained in the new version of the article, the measurements are phase-locked at the azimuth angle $60°$ after blade passage. Fig. 2 of the paper shows that the PIV windows were at the 3 o'clock position and the measurements were performed when one of the blades was located at the 11 o'clock position. As we also explain now in the manuscript, the available measurements were averaged over 31 image pairs. The individual image pairs are however not available, so reconstructing the time-history is unfortunately not possible.**

**References**

W. Z. Shen, H. MOL, and S. JN. Determination of angle of attack (aoa) for rotating blades. In Peinke, Schaumann, and Barth, editors, *Proceedings of the Euromech Colloquium – Wind Energy 2005*, pages 205–209, Oldenburg, Germany, 2006. Springer.

H. Yang, W. Z. Shen, J. N. Sørensen, and W. J. Zhu. Extraction of airfoil data using PIV and pressure measurements. *Wind Energy*, 14(4):539–556, 2011. ISSN 1099-1824. doi: 10.1002/we.441.

---

## Author Comment (AC4) · 15 Nov 2017

**Reply to the remarks of the editor to the comments of referee #2 on the manuscript "Brief communication: extraction of the wake induction and angle of attack on rotating wind turbine blades from PIV and CFD results"**

Iván Herráez, Elia Daniele and J. Gerard Schepers

**We would like to thank the editor for his complementary remarks to the comments of referee #2. In the following, we present our replies.**

**Comment**

1. *The comments of reviewer 2 are generally in line with those by reviewer 1. But he also states that the issue, if your method really represents the real AOA, needs to be supported by data, before the paper can be accepted for publication. Please address this plus the remaining items pointed out by the reviewer in your reply.*

**Reply:**
    **This issue is now extensively discussed in the paper and supported with data from references like Burton et al. [2011] as well as with an analytical derivation of the method in the appendix.**

**References**

T. Burton, D. Sharpe, N. Jenkins, and E. Bossanyi. *Wind Energy Handbook.* John Wiley & Sons, 2 edition, 2011.

---

## Referee Comment (RC3) · Anonymous Referee #1 · 16 Nov 2017

The authors have improved their paper and clearly described the advantages and disadvantages of their new method. Thus this reviewer recommends the paper for publication in WES.

---

## Editor Comment (EC3) · J.ÂăN. Sørensen (Editor) · 16 Nov 2017

I think that the authors have responded fully satisfactorily to the comments of the two reviewers. However, I need to have this approved by the reviewers. Please upload your reply to the rebuttal of the authors. Best Jens N. Sørensen Editor

---

## Author Response (AR1)

**Reply to the remarks of the editor on the manuscript "Extraction of the wake induction and angle of attack on rotating wind turbine blades from PIV and CFD results"**

Iván Herráez, Elia Daniele and J. Gerard Schepers

We would like to allude to the change of the title, where we have deleted "Brief communication" from it. The reason for this is that the manuscript does not fulfil the requirements for a brief communication (allowed are max. 2-4 pages and 3 figures, whereas the current version of the paper is 15 pages long and contains 6 figures). This change is highlighted in the attached version of the manuscript. No further changes have been performed.

[revised manuscript text omitted]